# Combined Treatment with Evogliptin and Temozolomide Alters miRNA Expression but Shows Limited Additive Effect on Glioma

**DOI:** 10.3390/ijms26199508

**Published:** 2025-09-28

**Authors:** Seung Yoon Song, Keun Soo Lee, Jung Eun Lee, Juwon Ahn, Jaejoon Lim, Seung Ho Yang

**Affiliations:** 1Department of Neurosurgery, St. Vincent’s Hospital, College of Medicine, The Catholic University of Korea, 93 Jungbu-daero, Paldal-gu, Suwon 16247, Republic of Korea; shforl86@naver.com (S.Y.S.); eunree@nate.com (J.E.L.); 2Department of Neurosurgery, Busan Paik Hospital, Inje University College of Medicine, 75 Bokji-ro, Busangin-gu, Busan 47392, Republic of Korea; auferstehung@hanmail.net; 3Department of Neurosurgery, Bundang CHA Medical Center, CHA University, 16 Yatap-ro 65beon-gil, Bundang-gu, Seongnam-si 13497, Republic of Korea; 72ysh@naver.com (J.A.); coolppeng@hanmail.net (J.L.)

**Keywords:** glioma, DPP4, evogliptin, temozolomide, miRNA

## Abstract

Dipeptidyl-peptidase IV (DPP4) inhibitors have shown potential anti-tumor properties. This study investigates the therapeutic potential of evogliptin, a DPP4 inhibitor, both as a single agent and in combination with temozolomide (TMZ), in glioma models. In vitro studies were performed using U87 and U373 glioma cell lines exposed to different concentrations of TMZ (250, 500 μM) and evogliptin (250, 500 ng/mL), either alone or together, for 24, 48, and 72 h. Cell viability was determined with the MTT assay. In vivo effectiveness was tested in a xenograft mouse model treated with intraperitoneal injections of evogliptin (60 mg/k g/day), TMZ (15 mg/kg/day), or their combination over 3 weeks. The combination of TMZ and evogliptin markedly reduced cell viability compared to single-agent treatments. DPP4 mRNA levels decreased more substantially with combination therapy. miRNA expression profiling with Affymetrix arrays indicated that certain miRNAs, such as miR-4440 and miR-6780b-5p, were upregulated after treatment with evogliptin or the combination regimen, whereas others were downregulated. These miRNAs could play a role in limiting glioma growth through DPP4 regulation. In the animal model, evogliptin alone did not provide a survival advantage. Analysis of TCGA data showed that glioma patients with decreased DPP4 expression had improved survival rates. The co-administration of evogliptin and temozolomide resulted in distinct miRNA profile changes. Nevertheless, both in vitro and in vivo, the added cytotoxicity from the combination was minimal.

## 1. Introduction

Gliomas represent some of the most aggressive primary central nervous system tumors, with glioblastoma (GBM) as the most common and malignant form. Despite improvements in multimodal therapy using extensive surgical resection, radiotherapy, and chemotherapy with temozolomide (TMZ), patient prognosis is still poor, showing a median survival of 12–15 months and a five-year survival rate below 10% [1,2]. Temozolomide, an oral alkylating chemotherapeutic, remains the principal agent in glioblastoma treatment; however, its benefit is limited by tumor heterogeneity and inherent resistance mechanisms [3]. Therefore, there is an increasing demand for combination therapies that can enhance the anti-tumor effect of TMZ and provide better clinical outcomes.

Recent studies have identified dipeptidyl peptidase IV (DPP4), a membrane-bound serine exopeptidase, as an important target in cancer research. DPP4 contributes to tumor progression by modulating immune responses, inflammation, and the remodeling of the extracellular matrix [4,5]. Evogliptin, a selective DPP4 inhibitor, is currently approved for managing type 2 diabetes mellitus. Preclinical evidence indicates that DPP4 inhibitors can exert anti-tumor effects, largely through their influence on the tumor microenvironment and attenuation of inflammatory pathways [6,7,8]. Elevated DPP4 expression correlates with worse clinical outcomes in several malignancies, such as gliomas [9]. Based on the growing evidence of DPP4’s role in glioma, it is relevant to assess evogliptin’s therapeutic potential in this context.

This study examines the anti-glioma properties of evogliptin as a single agent and in combination with TMZ, using both in vitro and in vivo models. Additionally, the study investigates mechanisms related to DPP4 protein expression and microRNA (miRNA) regulation, and assesses the clinical significance of DPP4 expression utilizing the Cancer Genome Atlas (TCGA) datasets.

## 2. Results

### 2.1. Cell Viability

To assess the cytotoxic impact of TMZ, evogliptin, and their potential combination, we performed MTT assays on U87 and U373 glioma cell lines following 72 h drug exposure. Live cells were identified using green fluorescent staining (Figure 1A). Two concentrations of each agent were tested to evaluate their individual cytotoxicity. TMZ produced a dose-dependent cytotoxic effect, with 500 μM causing significantly greater cell death than 250 μM. Evogliptin as a monotherapy reduced cell viability compared to the control; however, no obvious dose-dependent trend was noted between the 250 ng/mL and 500 ng/mL treatments, indicating a lesser cytotoxic effect with monotherapy.

The combination of TMZ with evogliptin resulted in further suppression of cell viability compared to TMZ alone in the low-dose TMZ (250 μM) group, particularly observed in U87 cells. In contrast, for the high-dose TMZ (500 μM) group, adding evogliptin did not yield a statistically significant additional decrease in cell viability, suggesting that the cytotoxic effect at this dose could be approaching its maximum, thereby providing minimal further benefit from the addition of evogliptin (Figure 1B). These findings indicate that TMZ and evogliptin combination did not produce a distinct additive effect in vitro.

### 2.2. DPP4 Expression

To evaluate the impact of treatment on DPP4 protein expression, we conducted Western blotting of U87 and U373 cells following 72 h exposure to TMZ, evogliptin, or their combination. In U87 cells, DPP4 expression levels were largely unaltered after monotherapy with evogliptin (500 ng/mL). Conversely, co-treatment with TMZ (500 μM) and evogliptin (500 ng/mL) produced a notable decrease in DPP4 expression, indicating a potential synergistic effect on DPP4 suppression at the protein level. A comparable trend was observed in the U373 cells. Single-agent treatment with either TMZ or evogliptin did not significantly modify DPP4 expression. In contrast, the combination resulted in decreased DPP4 protein levels, demonstrating that dual therapy is more effective at inhibiting DPP4 expression compared with monotherapy (Figure 1C).

These results suggest that the anti-tumor efficacy of the combination therapy may, at least in part, result from downregulating DPP4 expression in glioma cells.

### 2.3. U87 Xenograft Mice Models

U87 cells stably transduced with a firefly luciferase reporter were implanted intracranially into nude mice (*n* = 3 for control, *n* = 5 per treatment group). Bioluminescence imaging (BLI) was performed on days 10, 30, 35, and 46 after inoculation to monitor tumor growth (Figure 2A). By day 30, BLI signals had markedly increased in the control and evogliptin groups, with median survival times of 23 and 27 days, respectively (log-rank test, *p* = 0.008). Mice treated with TMZ or the TMZ + evogliptin combination showed reduced BLI signals on days 30 and 35, although signal reappeared by day 46. Median survival was not reached in either the TMZ or combination groups during the observation period (Figure 2B).

### 2.4. miRNA Analysis

To elucidate the molecular effects of evogliptin alone and in combination with TMZ, we performed miRNA microarray profiling using Affymetrix GeneChip miRNA 4.0 arrays in U87 and U373 cell lines. Evogliptin treatment (500 ng/mL) led to significant changes in miRNA expression levels, with 110 miRNAs upregulated and 48 downregulated in U87 cells, while 37 miRNAs were upregulated and 25 downregulated in U373 cells.

Among these, ten miRNAs were consistently upregulated in both cell lines: miR-1246, miR-149-3p, miR-3148, miR-4440, miR-4484, miR-4535, miR-6126, miR-6780b-5p, miR-6870-5p, and miR-7110-5p. Furthermore, eight miRNAs were consistently downregulated: miR-149-5p, miR-23a-5p, miR-3195, miR-4330, miR-4497, miR-4669, miR-671-3p, and miR-744-5p (Figure 3).

Of particular importance, miR-4440 and miR-6780b-5p displayed further upregulation following combination treatment with TMZ, indicating a potential synergistic effect on miRNAs implicated in glioma cell proliferation. These results indicate that evogliptin regulates unique miRNA profiles in glioma cells and that co-treatment with TMZ amplifies this regulatory influence. The miRNAs with altered expression may serve as critical mediators of the anti-tumor activity observed with the combination therapy (Appendix A).

### 2.5. TCGA Dataset and DPP4 Expression

To determine the clinical significance of DPP4 expression, we utilized Kaplan–Meier survival analysis on primary tumor samples from the TCGA cohort. High DPP4 expression was clearly linked to decreased OS (Figure 4). These data demonstrate that elevated DPP4 expression is strongly associated with poor prognosis, supporting the involvement of DPP4 in glioma progression and therapeutic response. Thus, DPP4 could represent both a valuable prognostic marker and an attractive therapeutic target for precision medicine in glioma (Appendix A).

## 3. Discussion

In this study, we examined the in vitro cytotoxic potential of combining the DPP4 inhibitor evogliptin with the standard chemotherapeutic agent TMZ in glioma cell lines. Although TMZ treatment alone produced a significant, dose-dependent decrease in cell viability, evogliptin as a single agent showed only minimal efficacy and lacked a clear dose–response effect. Combined treatment with TMZ and evogliptin failed to demonstrate a uniformly additive effect in vitro and in vivo. However, combination therapy led to the suppression of DPP4 protein expression and altered the expression of specific miRNAs implicated in tumor progression. These findings suggest that DPP4 inhibition may enhance the therapeutic effect of TMZ through diverse molecular mechanisms and supports the development of this novel combinatorial approach for glioma management. In our study, we did not observe a clear synergistic effect between evogliptin and temozolomide. Rather, the combination produced at most modest additive changes, suggesting that the observed DPP4 protein reduction may reflect stress-related turnover or post-transcriptional regulation rather than true pharmacologic synergy.

TMZ, an alkylating agent that induces DNA damage and oxidative stress responses, may increase the sensitivity of glioma cells to DPP4 inhibition, either by altering transcriptional regulators of the DPP4 gene or by modifying the post-translational stability of the protein. It is also conceivable that TMZ-induced cellular stress augments proteasome-mediated degradation of DPP4 or influences upstream epigenetic or microRNA networks regulating DPP4 expression [10,11]. Importantly, DPP4 inhibitors such as evogliptin are primarily developed to diminish enzymatic activity rather than directly decreasing protein abundance [12,13]. Consequently, the observed reduction in DPP4 protein levels when combining therapies indicates a distinct synergistic effect, wherein TMZ-induced cellular stress (such as DNA damage or oxidative signaling) may potentiate evogliptin’s capacity to inhibit DPP4 expression by impacting transcription or facilitating protein degradation. Prior studies have demonstrated that chemotherapeutic agents can affect membrane-associated enzymes like DPP4 by altering cytokine profiles, redox homeostasis, or NF-κB signaling pathways [14,15], which are closely related to the pharmacological actions of TMZ.

The miRNA profiling additionally demonstrated that evogliptin monotherapy resulted in upregulation of multiple tumor-suppressive miRNAs, including miR-149-3p and miR-3148, both recognized for their roles in suppressing glioma cell proliferation and invasion. Notably, miR-149 inhibits glioma progression by targeting AKT1, thereby attenuating the PI3K/AKT signaling cascade—a critical pathway in glioma cell survival and therapy resistance [16]. Likewise, miR-3148 has been associated with inhibiting the proliferation and migration of glioma cells [17]. In comparison, overexpression of miR-23a has been shown to enhance glioma cell invasion, likely through modulation of MMP-14 by directly repressing HOXD10 expression, which has recently been identified as a target of miR-23a [18]. Several other deregulated miRNAs identified in our dataset have been previously implicated in glioma biology. For instance, miR-1246 has been reported to promote tumor cell proliferation and angiogenesis via the STAT3/VEGF pathway [19], whereas miR-149-3p acts as a tumor suppressor by targeting CDK6 and MMP2, thereby influencing invasion and cell cycle progression [20]. miR-23a-5p is known to may function as a tumor suppressor miRNA by regulating chromobox 2/Wnt/β-catenin pathway, and miR-4440 has been associated with glioma stemness and epithelial–mesenchymal transition [21]. The downregulation of these miRNAs by evogliptin points to a shift in the transcriptomic profile toward a phenotype characterized by reduced malignancy and increased sensitivity to therapy.

Importantly, the combination of evogliptin and TMZ further amplified this regulatory pattern by synergistically upregulating miR-4440 and miR-6780b-5p, which were only minimally influenced when either agent was used as monotherapy. Although the precise functions of these miRNAs in glioma remain incompletely understood, bioinformatic analysis and recent studies indicate that they may target DPP4 mRNA or associated downstream pathways, thereby mediating DPP4 downregulation and restricting tumor-promoting mechanisms [22,23]. This dual targeting approach—modulating both protein expression (DPP4) and regulatory non-coding RNAs (miRNAs) in parallel—illustrates a layered molecular reprogramming effect, ultimately enhancing the cytotoxic and anti-proliferative outcomes beyond what is achievable with either agent alone. Future studies should validate miRNA-DPP4 targeting (e.g., luciferase assays) and functional impacts (e.g., transfection/knockdown).

Altogether, these findings support the view that evogliptin functions not only as a biochemical inhibitor of DPP4 but also as a transcriptomic modulator, reshaping the tumor’s regulatory network [24,25]. In combination with TMZ, the resulting epigenetic and post-transcriptional synergy may be responsible for the observed increase in glioma cell death and greater therapeutic responsiveness. However, our results indicate that these did not translate to substantial additive therapeutic effects in vivo, while combination therapy induced significant molecular changes. Additionally, analysis of TCGA datasets indicated that lower DPP4 expression is associated with improved survival. This discrepancy may be due to several factors: compensatory signaling pathways that limit the impact of DPP4 inhibition alone, restricted blood–brain barrier penetration of evogliptin, tumor microenvironment differences between the preclinical model and human glioma biology, and limitations of small-scale animal experiments compared to large clinical datasets.

In line with our findings on evogliptin, recent studies have also reported antitumor effects of other DPP4 inhibitors in glioma models. For example, linagliptin suppressed cell viability, proliferation, and migration of U87, U251, A172, and GL261 cells, and significantly reduced tumor progression in a murine GBM model. Furthermore, linagliptin was shown to disrupt the DPP4–EGFR interaction and synergize with cPLA2 inhibition or temozolomide to enhance therapeutic efficacy in glioma [26].

These findings reinforce the clinical significance of DPP4 both as a therapeutic target and as a prognostic biomarker in glioma. Despite these encouraging results, several limitations should be considered. First, the link between DPP4 inhibition and miRNA modulation is correlative and has not been shown to be causal; functional experiments employing miRNA overexpression or knockdown are required to establish mechanistic insights. Second, although U87 and U373 cell lines are standard models for glioma research, they cannot fully represent the molecular and phenotypic diversity of patient-derived gliomas. Future studies should include glioma stem-like cells or patient-derived organoids to better recapitulate clinical features. Third, while DPP4 protein expression was analyzed, direct assessment of enzymatic function and impacts on the immune microenvironment would provide a more comprehensive understanding of the mechanism. Moreover, further research is needed to characterize the effects of evogliptin on blood–brain barrier permeability, immune cell infiltration, and glioma metabolic pathways. Fourth, we did not perform systematic dose–response evaluations of evogliptin nor formal synergy analyses. In addition, the use of Affymetrix arrays, while informative, provides narrower coverage than bulk RNA sequencing and does not fully capture in vivo tumor–host interactions.

## 4. Material and Methods

### 4.1. Cell Culture

Human glioma cell lines U87-HTB14 and U373 were purchased from the American Type Culture Collection (ATCC, Manassas, VA, USA). After completing cell authentication (STR profiling) and mycoplasma negativity confirmation, cells were maintained in high-glucose DM (Thermo Fisher Scientific, Waltham, MA, USA) supplemented with 10% FBS (Gibco 16000-044) and 1% penicillin-streptomycin (Gibco 15140-122) at 37 °C in a 5% CO_2_ incubator. TMZ (Sigma Aldrich, Merck KGaA, Darmstadt, Germany) was dissolved in dimethyl sulfoxide (DMSO) to obtain a 200 mM stock solution, then diluted in cell culture medium for experimental use.

### 4.2. Fluorescence Imaging

After 72 h of drug treatment, cells were washed twice with PBS, mounted with coverslips, and observed using a fluorescence microscope (LSM900w/AiryscanII, Carl Zeiss, Oberkochen, Germany). GFP signals were acquired at excitation 470–490 nm/emission 510–550 nm (FITC channel). The same exposure time and gain were applied to all groups. ≥5 fields of view per well were imaged, and the scale bar was set to 200 µm.

### 4.3. Cell Viability Assay

U87 and U373 (1 × 10^4^ cells/well) were seeded into 96-well flat-bottom tissue culture plates and maintained at 37 °C in a humidified atmosphere of 5% CO_2_ and 95% air. The cells were exposed for 24, 48, and 72 h to either 250 or 500 μM TMZ, or to 250 or 500 ng/mL evogliptin for the same durations. Subsequently, combined treatment with TMZ and evogliptin was conducted following the same protocol. At each time point (24, 48, and 72 h), 10 μL of MTT [3-(4,5-dimethylthiazol-2-yl)-2,5-diphenyltetrazolium bromide] stock solution (Ez-CyTox, Daeil Lab Service Co., Ltd., Seoul, Republic of Korea) was added per well, and the plates were incubated according to the manufacturer’s instructions. The resulting formazan was solubilized, and absorbance was measured at 570 nm with a reference wavelength of 630 nm using a microplate reader.

Cell viability was calculated as the percentage of absorbance relative to untreated controls. All experiments were performed in five independent replicates (n = 5). Data are expressed as mean ± SEM, and statistical analyses were performed using GraphPad Prism 8 (GraphPad Software 8.0). Group comparisons were conducted by one-way ANOVA followed by Tukey’s post hoc test, with *p* < 0.05 considered statistically significant.

### 4.4. DPP4 Western Blot

U87 and U373 glioma cells were exposed to temozolomide (500 μM), evogliptin (500 ng/mL), or their combination for 72 h. Following treatment, the cells were lysed in RIPA buffer supplemented with protease inhibitors, and total protein amounts were quantified using the BCA assay. Equal amounts of extracted protein were resolved using SDS-PAGE, transferred onto PVDF membranes, and blocked in 5% non-fat milk. The membranes were incubated overnight at 4 °C with anti-DPP4 (1:1000, Abcam, Cambridge, UK) and anti-β-actin (1:5000, Sigma-Aldrich) antibodies. After incubation with HRP-conjugated secondary antibodies, protein bands were detected by enhanced chemiluminescence and analyzed with ImageJ software, version 1.53k. DPP4 protein levels were normalized to β-actin expression.

### 4.5. Intracranial Inoculation of Cancer Cells and Experimental Design

The study protocol was reviewed and approved by the Institutional Review Board of St. Vincent’s Hospital, The Catholic University of Korea (IACUC 20-04). Athymic nude mice were anesthetized by intraperitoneal (i.p.) injection of 12 mg/kg xylazine (Rompun; Cutter Laboratories, Shawnee, KS, USA) and 30 mg/kg ketamine (Ketalar; Parke-Davis & Co., Morris Plains, NJ, USA). Mice were placed in a stereotactic frame, and 5 × 10^5^ U87 cells (200 µL) were injected into the right frontal lobe (2 mm lateral and 1 mm anterior to bregma, at a depth of 2.5 mm from the skull surface) using a sterile Hamilton syringe with a 26-gauge needle (Hamilton Co., Reno, NV, USA) and a microinfusion pump (Harvard Apparatus, Holliston, MA, USA).

Mice in the first treatment group received evogliptin (60 mg/kg/day) orally for 3 weeks following intracranial tumor inoculation with U87 cells. The second treatment group was administered TMZ (15 mg/kg/day) by i.p. injection for 3 weeks post-inoculation. In the combination treatment groups, mice were given evogliptin (60 mg/kg/day) by oral route and TMZ (15 mg/kg/day) by i.p. injection for 3 weeks. Mice were euthanized approximately 8 weeks after the stereotactic injection. The criterion for determining the endpoint was a 20% loss of body weight, as visual assessment of tumor size was not feasible; however, mice were euthanized if they exhibited an inability to carry out normal physiological activities such as eating, defecating, and urinating. The euthanasia procedure for mice was conducted in a dedicated acrylic chamber using CO_2_ gas. The filling rate was maintained at 30–70% per min to ensure a gradual and controlled process. Following an approximate exposure time of 5 min to 100% CO_2_ within the chamber, the gas flow was continued for an additional minimum of 1 min. This duration allowed for the confirmation of definitive veterinary death, specifically by observing evident signs such as respiratory arrest. The employed protocol aimed to ensure a humane and effective euthanasia procedure. The animal studies detailed here took place between the years 2020 and 2022. IACUC and Department of Laboratory Animal in The Catholic University of Korea accredited the Korea Excellence Animal Laboratory Facility from Korea Food and Drug Administration in 2017 and acquired AAALAC International full accreditation in 2018.

### 4.6. Affymetrix miRNA Arrays

For quality control, RNA purity and integrity were assessed using the OD 260/280 ratio and further examined with the Agilent 2100 Bioanalyzer (Agilent Technologies, Palo Alto, CA, USA). The Affymetrix Genechip miRNA 4.0 array protocol was conducted following the manufacturer’s instructions. In detail, 1 µg of RNA from each sample was labeled using the FlashTag™ Biotin RNA Labeling Kit (Genisphere, Hatfield, PA, USA). Quantification, fractionation, and hybridization of labeled RNA to the miRNA microarray were all carried out in accordance with the manufacturer’s standard protocols. The labeled RNA samples were incubated at 99 °C for 5 min, followed by incubation at 45 °C for 5 min. Hybridization with the RNA array was performed with continuous agitation at 60 rotations per minute for 16 h at 48 °C on the Affymetrix^®^ 450 Fluidics Station. Washing and staining of the chips were performed on a Genechip Fluidics Station 450 (Affymetrix, Santa Clara, CA, USA). The chips were scanned using an Affymetrix GCS 3000 scanner (Affymetrix, Santa Clara, CA, USA). Signal intensities were calculated using the Affymetrix^®^ GeneChip™ Command Console software. Putative target genes of the differentially expressed miRNAs were predicted using established bioinformatic algorithms, including TargetScan (v7.2), miRDB, and miRTarBase. Only targets predicted by at least two of these databases were considered for downstream analysis. Differentially expressed miRNAs were defined as those showing an absolute fold-change ≥ 2.0 with an adjusted *p* value < 0.05.

### 4.7. TCGA Data Acquisition, Processing and Survival Analysis

Transcriptomic and clinical data for glioma patients were obtained from the UCSC Xena platform (https://xenabrowser.net/datapages/, accessed on 4 June 2023). The dataset included normalized RNA sequencing data (RNAseqv2 RSEM), as well as associated clinical and survival outcome data. Gene-level transcription estimates from the RNA-seq data were expressed as log2(x + 1) transformed RSEM normalized counts. In total, 694 glioma specimens were analyzed in this study. Analyses were performed to determine the relationship between dipeptidyl peptidase-4 (DPP4) expression and patient survival using this cohort. Differences in prognosis related to DPP4 expression levels were evaluated independently. All data preprocessing and subsequent analyses were conducted with tools provided by the UCSC Xena online platform.

All statistical and survival analyses were executed using R software version 4.2.3. For comparisons of survival between clusters defined by agglomerative hierarchical clustering of complete gene expression profiles, pairwise log-rank tests were used, with *p*-values adjusted for multiple comparisons via the Benjamini–Hochberg procedure. Kaplan–Meier plots, generated using the survminer R package, were used to visualize survival distributions. Analyses included both the Kaplan–Meier method and Cox proportional hazards regression for survival modeling. Patients who were alive at the date of last contact were classified as censored. Hazard ratios (HRs) and corresponding 95% confidence intervals (CIs) were calculated to assess variable effects on survival, and statistical significance was established at a *p*-value threshold of less than 0.05.

### 4.8. Statistical Analysis

All quantitative data from cell-based assays were statistically analyzed using two-tailed Student’s *t*-tests. Kaplan–Meier survival curves were constructed, and the log-rank test was utilized to assess differences between groups. Statistical significance was defined as a *p*-value less than 0.05. All statistical analyses were conducted with SPSS (version 26.0; IBM Corp., Armonk, NY, USA).

## 5. Conclusions

Our findings indicate that evogliptin, especially in combination with TMZ, may participate in dual mechanisms that include suppression of DPP4 expression and regulation of tumor-associated miRNAs. Subsequent studies should prioritize confirming these mechanisms in clinically relevant models and further investigate the wider immunometabolic consequences of DPP4 inhibition within the glioma microenvironment.

## Figures and Tables

**Figure 1 ijms-26-09508-f001:**
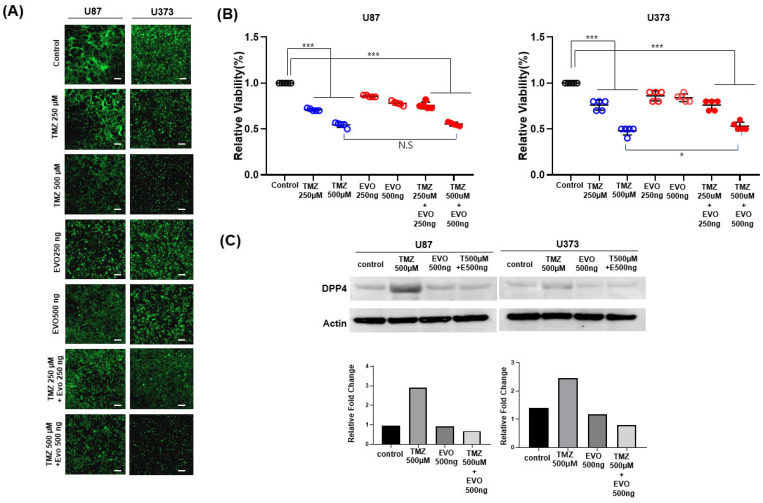
Impact of TMZ and evogliptin treatments on cell viability and DPP4 expression in the glioblastoma cell lines U87 and U373. (**A**) Representative fluorescence microscopy images stained with live cell markers following 72 h treatment with TMZ (250 μM or 500 μM), evogliptin (250 ng/mL or 500 ng/mL), or their combination (scale bar = 200 μm). (**B**) Quantitative analysis of the relative fold change in cell viability is displayed in the lower panels (*n* = 6 per group). TMZ treatment resulted in a dose-dependent reduction in viability, whereas evogliptin alone produced limited effects without a clear dose-dependent response. Combination therapy induced a slight increase in cytotoxicity at low-dose TMZ, but did not yield a significant additive effect at high-dose TMZ. Data are presented as the mean ± SEM. Statistical assessments were performed using one-way ANOVA followed by post hoc analysis (* *p*  <  0.05, *** *p*  <  0.001, NS: not significant). (**C**) Western blot analysis for DPP4 protein expression. Cells received TMZ (500 μM), evogliptin (500 ng/mL), or their combination for 72 h. In both U87 and U373 cell lines, single-agent TMZ or evogliptin treatment resulted in minimal alterations in DPP4 protein levels. However, the combination of TMZ and evogliptin distinctly reduced DPP4 expression, suggesting a synergistic suppressive effect on DPP4.

**Figure 2 ijms-26-09508-f002:**
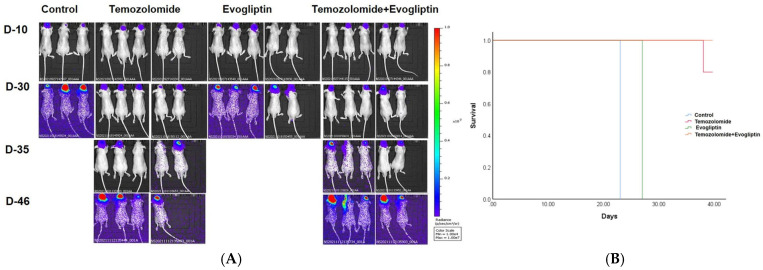
Bioluminescence imaging (BLI) and survival curve analysis in xenograft mouse models. (**A**) BLI was conducted on days 10, 30, 35, and 46 after tumor cell inoculation to assess tumor growth over time. (**B**) Kaplan–Meier survival curves for each group. Control and evogliptin groups had median survivals of 23 and 27 days, respectively (*p* = 0.008). Median survival was not reached in the TMZ and combination treatment groups during the observation period. Survival was analyzed by the log-rank test.

**Figure 3 ijms-26-09508-f003:**
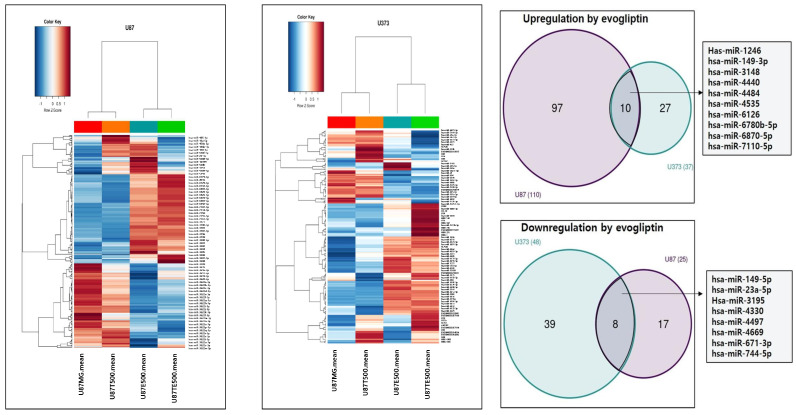
Heatmap representation of differentially expressed miRNAs in U87 and U373 cells exposed to evogliptin. Ten miRNAs, including miR-1246, miR-149-3p, miR-3148, and miR-4440, were found to be upregulated in both cell types. Conversely, eight miRNAs such as miR-23a-5p and miR-744-5p showed consistent downregulation. These miRNAs may serve as potential mediators of the anti-tumor effects exerted by evogliptin.

**Figure 4 ijms-26-09508-f004:**
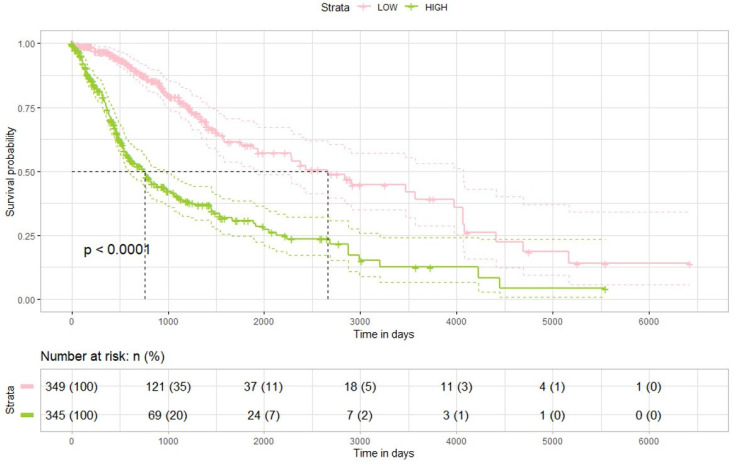
Kaplan–Meier survival analysis of glioma patients from the TCGA dataset, stratified by DPP4 expression levels. Patients with low DPP4 expression demonstrated significantly improved overall survival compared to those with high DPP4 expression (log-rank *p* < 0.05). The dashed lines denote the median survival time for each group.

## Data Availability

The datasets produced in this study can be obtained from the corresponding author upon reasonable request.

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
