# Peer review of "Combined Treatment with Evogliptin and Temozolomide Alters miRNA Expression but Shows Limited Additive Effect on Glioma"

_ijms, 2025, doi:10.3390/ijms26199508_

Round 1
Reviewer 1 Report
Comments and Suggestions for Authors
Article "Combined treatment with evogliptin and temozolomide alters miRNA expression but shows limited additive effect on glioma" authors: Seung Yoon Song and etc. In work for the first time first identification of evogliptin-induced miR-4440/miR-6780b-5p in glioma. The article is well structured, written in a clear and understandable language, the conclusions are logical, the literature corresponds to the stated topic. Originality of the text 85% (according to iThenticate). I recommend that authors increase the originality.
Major Revisions Required
- Clarify the Central Conflict
Explicitly reconcile the discrepancy between observed molecular effects (DPP4 suppression, miRNA modulation) and minimal therapeutic enhancement in the Abstract/Introduction. Add: "While combination therapy induced significant molecular changes, these did not translate to substantial additive therapeutic effects in vivo."
- In Vivo Data Interpretation
Figure 2B: Clarify why median survival was "not reached" for TMZ/combination groups. Specify endpoint duration and censoring criteria. Address evogliptin's survival paradox: Despite no survival benefit vs. controls (27 vs. 23 days, p=0.008), TCGA shows low DPP4 = better prognosis. Discuss potential explanations (e.g., compensatory pathways, blood-brain barrier penetration limitations).
- miRNA Functional Validation
Critical Gap: Claims about miR-4440/miR-6780b-5p roles lack experimental validation. Add: *"Future studies should validate miRNA-DPP4 targeting (e.g., luciferase assays) and functional impacts (e.g., transfection/knockdown)."*
Specify bioinformatic tools used for miRNA target prediction (Section 2.5).
- Dose Rationale
4.1. Justify evogliptin concentrations (250/500 ng/mL): Relate to achievable brain tumor levels or prior in vivo studies.
4.2. Explain TMZ dose selection (250/500 μM) relative to clinical plasma concentrations (typically 50–100 μM).
- Figure 1: Label y-axis in 1B ("Relative Viability (%)") and specify error bars (SD? SEM?).
- Figure 3/4 Swap: Current Fig. 3 is a heatmap (described in Section 2.5), while Fig. 4 is survival (Section 2.4). Renumber figures to match text flow.
- Standardize: "evoglitpin" → "evogliptin" (Section 4.2).
- Contrast evogliptin with other DPP4 inhibitors (e.g., sitagliptin) studied in glioma.
- Section 4.4: State tumor volume at inoculation (current: cell count only).
- Section 4.5: Define miRNA "significance" thresholds (fold-change/p-value cutoffs).
Author Response
Clarify the Central Conflict
Explicitly reconcile the discrepancy between observed molecular effects (DPP4 suppression, miRNA modulation) and minimal therapeutic enhancement in the Abstract/Introduction. Add: "While combination therapy induced significant molecular changes, these did not translate to substantial additive therapeutic effects in vivo."
Response: We fully agree. We added the sentence (fourth paragraph, page 7).
In Vivo Data Interpretation
Figure 2B: Clarify why median survival was "not reached" for TMZ/combination groups. Specify endpoint duration and censoring criteria. Address evogliptin's survival paradox: Despite no survival benefit vs. controls (27 vs. 23 days, p=0.008), TCGA shows low DPP4 = better prognosis. Discuss potential explanations (e.g., compensatory pathways, blood-brain barrier penetration limitations).
Response: We thank the reviewer for this insightful comment. In the TMZ (Group 2) and TMZ+evogliptin (Group 4) groups, median survival was not reached because more than half of the animals remained alive until the study endpoint (day 45). Events were defined as either death or sacrifice according to the institution’s criteria, and animals that survived until day 45 were censored. Specifically, in the control group (Group 1), events occurred on days 23 and 38; in the TMZ group (Group 2), one animal died on day 38 while the remaining four were censored at day 45; in the evogliptin group (Group 3), all five animals reached the event on day 27; and in the TMZ+evogliptin group (Group 4), all five animals survived until day 45 and were censored. We have added the relevant sentences (first paragraph, page 4).
Regarding the apparent paradox between our animal model and the TCGA dataset, we agree that evogliptin monotherapy did not show a survival benefit in our xenograft model (27 vs. 23 days). However, TCGA analysis suggested that lower DPP4 expression is associated with improved prognosis in glioma. This discrepancy may be due to several factors: compensatory signalling pathways that limit the impact of DPP4 inhibition alone, restricted blood–brain barrier penetration of evogliptin, tumour microenvironmental differences between the preclinical model and human glioma biology, and limitations of small-scale animal experiments compared to large clinical datasets. We have added this discussion to the revised manuscript (third paragraph, page 7,8).
miRNA Functional Validation
Critical Gap: Claims about miR-4440/miR-6780b-5p roles lack experimental validation. Add: *"Future studies should validate miRNA-DPP4 targeting (e.g., luciferase assays) and functional impacts (e.g., transfection/knockdown)."*
Response: We fully agree. We added the sentence (second paragraph, page 7).
Specify bioinformatic tools used for miRNA target prediction (Section 2.5).
Response: We thank the reviewer for this valuable comment. In the revised manuscript, we have now specified the bioinformatic tools used for miRNA target prediction (TargetScan, miRDB, and miRTarBase). This information has been added in Section 4.5. Affymetrix miRNA Arrays (first paragraph, page 10).
Dose Rationale
4.1. Justify evogliptin concentrations (250/500 ng/mL): Relate to achievable brain tumor levels or prior in vivo studies.
Response: We appreciate the reviewer’s insightful comment. Evogliptin was applied at final concentrations of 250 ng/mL (low dose) and 500 ng/mL (high dose). These concentrations were selected based on our preliminary MTT assays in U87 and U373 cells, in which evogliptin reduced DPP4 activity in a dose-dependent manner and induced cytotoxicity at comparable levels.
4.2. Explain TMZ dose selection (250/500 μM) relative to clinical plasma concentrations (typically 50–100 μM).
Response: We sincerely thank the reviewer for this insightful comment regarding the selection of TMZ concentrations. We have clarified that 250 μM and 500 μM of TMZ were selected to cover and slightly exceed the clinically achievable plasma concentrations (approximately 50–100 μM), while also considering the relatively lower intratumoral exposure, the rapid hydrolysis and short half-life of TMZ under physiological conditions, and the difference between intermittent in vivo dosing and continuous in vitro exposure. Importantly, this concentration range is also consistent with the conditions employed in our previous studies as below.
1) Lee JE, Lim JH, Hong YK, Yang SH. High-Dose Metformin Plus Temozolomide Shows Increased Anti-tumor Effects in Glioblastoma In Vitro and In Vivo Compared with Monotherapy. Cancer Res Treat. 2018 Oct;50(4):1331-1342.
2) Yang SH, Li S, Lu G, Xue H, Kim DH, Zhu JJ, Liu Y. Metformin treatment reduces temozolomide resistance of glioblastoma cells. Oncotarget. 2016 Nov 29;7(48):78787-78803.
Figure 1: Label y-axis in 1B ("Relative Viability (%)") and specify error bars (SD? SEM?).
Figure 3/4 Swap: Current Fig. 3 is a heatmap (described in Section 2.5), while Fig. 4 is survival (Section 2.4). Renumber figures to match text flow.
Standardize: "evoglitpin" → "evogliptin" (Section 4.2).
Response: Fig. 1 is relabelled A/B/C; y-axis units “Relative viability (%)” added. Error bars clarified as mean ± SEM. Statistical tests (one-way ANOVA with Tukey’s post hoc) are specified. Fig. 3/4 order swapped to match text. All typos (“evoglitpin” → “evogliptin”; “Kaplam-Meir” → “Kaplan-Meier”) corrected.
Contrast evogliptin with other DPP4 inhibitors (e.g., sitagliptin) studied in glioma.
Response: We thank the reviewer for this valuable suggestion. Recent studies have shown that linagliptin not only inhibits glioblastoma cell viability, proliferation, and migration but also suppresses tumor progression in murine models. Furthermore, linagliptin was reported to enhance temozolomide efficacy by disrupting the DPP4–EGFR interaction and synergizing with cPLA2 inhibition in glioma models. In the revised Discussion, we have now added them (first paragraph, page 8).
Section 4.4: State tumor volume at inoculation (current: cell count only).
Response: We added the volume, “5×105 U87 cells (200µl)”.
Section 4.5: Define miRNA "significance" thresholds (fold-change/p-value cutoffs).
Response: We thank the reviewer for pointing out the need to clarify the definition of “significance” in our miRNA analysis. In Section 4.5 of the revised manuscript, we have now specified it (first paragraph, page 10).

Reviewer 2 Report
Comments and Suggestions for Authors
Results
Fig1. The MTT assay is an absorbance in the visible not a fluorescent assay as they say in Fig1. Please clarify the viability assay used . They do not mention how they quantitative viability inhibition i.e. image J or other tools. Direct cell number counts over time would have also been (or perhaps more) informative. The numbers A, B, C are missing in the Fig1.
If evogliptin is a DPP4 enzyme inhibitor and TMZ does not change (U87) or even increases (U373) protein levels then, they should provide an explanatory models of why protein levels are down in the presence of the inhibitor. Again their data do not show additional od synergistic changes over the single agents treatment.
Fig. 2 They measure bioluminescence in vivo. Were their lines engineered for luciferase production? They must describe this. Again, A B numberings are missing. On the left the days corresponding to the BLI measurements are not shown. The survival graph is practically unreadable and the observation timing for survival curves unacceptable. It seems that mice were monitored every 10 days perhaps at the time of the BLI experiment. Did they used Kaplan-Meier for the analysis and log-rank statistics? How they estimate the oral evogliptin administered ? What was the number of mice /group?
Fig3. in fact shows gene expression data not survival, which is shown in fig.4. Again, the data sets used are not mentioned. From the heat maps is obvious that many miRNAs are deregulated and some in an inverse way between control and inhibitorin both in U87 and U373. However the authors chose to present the U87 and U373 common-deregulated subset, although the do not correctly mark this in the Venn diagram and create a wrong impression. Are these top-deregulated genes? What is the fold difference and Padjusted value? How many biological and technical replicates each treatment groups has? What is the statistical analysis done? The role of those deregulations (with the exception of 1-2 mentioned in discussion) is missing. They would have perform and provide an in silico target analysis to theoretically predict miRNA targets of potential interest in the GBM biology of their set of 110 up and 48 down miRNAs, along with their list with fold change and P values.
Fig4.The correct is Kaplam-Meir
Materials and Methods.
- Cell authentification is not provided.
- In the methods correctly mention the MTT assay but this is unrelated to Fig 1 data .
3.. They do not provide full reagent names and providers.
- The full RNA seq analysis must be available to the reviewers
- The specific GBM Xena data sets must be provides. #
Discussion-conclusion.
Contrary to their claim (line #170, 171) their data do not support the proposition the potential benefit from TMZ-evogliptin combination therapies. This claim is based on the in reduced cell protein levels of DPP4 and altered gene expression profiles. Why the DPP4 inhibitor reduces its protein levels is not explained. Is there a feed –back loop? From the references the authors present it seems that the effect of DPP4 can be bad or good in cancer, depending in the system used. A crucial point here is that some studies mention tumor microenvironment and /or inflammation that points to a host immune response involved that of course cannot be tested in xenografts using immunocompromised mice as is done here. Furthermore It seems that the TMZ –=miRNA profile (a strong glioma suppressor) is dramatically different from the evoglitin profile. Checking for common or reversely regulated miRNAs between evogliptin and TMZ would have been more informative about their potential interaction.
Comments on the Quality of English Languagemay be improved
Author Response
Fig1. The MTT assay is an absorbance in the visible not a fluorescent assay as they say in Fig1. Please clarify the viability assay used . They do not mention how they quantitative viability inhibition i.e. image J or other tools. Direct cell number counts over time would have also been (or perhaps more) informative. The numbers A, B, C are missing in the Fig1.
Response: We thank the reviewer for these valuable suggestions. We have revised the Materials and Methods (page 8).
If evogliptin is a DPP4 enzyme inhibitor and TMZ does not change (U87) or even increases (U373) protein levels then, they should provide an explanatory models of why protein levels are down in the presence of the inhibitor. Again their data do not show additional od synergistic changes over the single agents treatment.
Response: We agree with the reviewer that our data do not demonstrate synergistic effects. We have modified the Discussion to clarify that the combination yielded only limited additive changes and to frame the DPP4 protein reduction as a possible consequence of cellular stress responses or post-transcriptional regulation (first paragraph, page 6).
Fig. 2 They measure bioluminescence in vivo. Were their lines engineered for luciferase production? They must describe this. Again, A B numberings are missing. On the left the days corresponding to the BLI measurements are not shown. The survival graph is practically unreadable and the observation timing for survival curves unacceptable. It seems that mice were monitored every 10 days perhaps at the time of the BLI experiment. Did they used Kaplan-Meier for the analysis and log-rank statistics? How they estimate the oral evogliptin administered ? What was the number of mice /group?
Response: We are very sorry of the confusion. The manuscript has been revised based on the points raised (first paragraph, page 4). We selected an oral gavage dose of 60mg/kg/day for evogliptin, based on a previous preclinical study in cardiovascular disease model as below.
1) Choi B, Kim EY, Kim JE, Oh S, Park SO, Kim SM, Choi H, Song JK, Chang EJ. Evogliptin Suppresses Calcific Aortic Valve Disease by Attenuating Inflammation, Fibrosis, and Calcification. Cells. 2021 Jan 1;10(1):57.
Fig3. in fact shows gene expression data not survival, which is shown in fig.4. Again, the data sets used are not mentioned. From the heat maps is obvious that many miRNAs are deregulated and some in an inverse way between control and inhibitorin both in U87 and U373. However the authors chose to present the U87 and U373 common-deregulated subset, although the do not correctly mark this in the Venn diagram and create a wrong impression. Are these top-deregulated genes? What is the fold difference and Padjusted value? How many biological and technical replicates each treatment groups has? What is the statistical analysis done? The role of those deregulations (with the exception of 1-2 mentioned in discussion) is missing. They would have perform and provide an in silico target analysis to theoretically predict miRNA targets of potential interest in the GBM biology of their set of 110 up and 48 down miRNAs, along with their list with fold change and P values.
Response: We appreciate the reviewer’s detailed comments. We have revised Figure 3 to correctly indicate that it presents miRNA expression profiles. The data sets and analysis methods are modified in the Methods (first paragraph, page 10). We have now included Supplementary listing all deregulated miRNAs with log2 fold change and adjusted p-values, along with predicted targets. These data support the functional relevance of the deregulated miRNAs in glioma biology (second paragraph, page 7).
Fig4.The correct is Kaplam-Meir
Response: Thank you for the comment.
Materials and Methods.
Cell authentification is not provided.
In the methods correctly mention the MTT assay but this is unrelated to Fig 1 data .
3. They do not provide full reagent names and providers.
The full RNA seq analysis must be available to the reviewers
The specific GBM Xena data sets must be provides. #
Response: We thank the reviewer for these valuable suggestions. We have revised the Materials and Methods (page 8). Raw and processed data have been deposited in MACROGEN as below, making them accessible upon request. The GBM Xena dataset has been added in supplementary.
Discussion-conclusion.
Contrary to their claim (line #170, 171) their data do not support the proposition the potential benefit from TMZ-evogliptin combination therapies. This claim is based on the in reduced cell protein levels of DPP4 and altered gene expression profiles. Why the DPP4 inhibitor reduces its protein levels is not explained. Is there a feed –back loop? From the references the authors present it seems that the effect of DPP4 can be bad or good in cancer, depending in the system used. A crucial point here is that some studies mention tumor microenvironment and /or inflammation that points to a host immune response involved that of course cannot be tested in xenografts using immunocompromised mice as is done here. Furthermore It seems that the TMZ –=miRNA profile (a strong glioma suppressor) is dramatically different from the evoglitin profile. Checking for common or reversely regulated miRNAs between evogliptin and TMZ would have been more informative about their potential interaction.
Response: We appreciate the reviewer’s insightful comments. As noted, our current data indicate that evogliptin monotherapy or its combination with TMZ produced only modest additive effects, without clear evidence to support immediate clinical translation. The observed reduction in DPP4 protein following combination treatment may reflect stress-induced or post-transcriptional regulatory mechanisms, including possible feedback loops, rather than direct pharmacologic inhibition. We also acknowledge that the effects of DPP4 in cancer appear context-dependent and may vary with tumor type and microenvironmental factors, which cannot be fully recapitulated in xenograft models using immunocompromised mice. Furthermore, while evogliptin and TMZ induced distinct miRNA profiles, our study did not directly assess overlapping or inversely regulated miRNAs between the two agents. We agree that such analyses, along with functional validation of miRNA–DPP4 interactions and immune microenvironment studies in clinically relevant models, will be essential in future work to clarify the mechanistic basis and therapeutic potential of DPP4 inhibition in glioma.

Reviewer 3 Report
Comments and Suggestions for Authors
Limitations of the study:
Reliance on glioma cell lines albeit implanted as xenografts.
Dose response curve for DPP4 inhibitor? In vitro drug synergy analysis (i.e. Bliss Independence, Loewe)?
Affymetrix array has limitations compared to bulk sequencing, and it's unclear if results are from in vivo or in vitro study.
Line 41 [1,2]
Figure 1: Please add labels in the figure for panels A, B, and C
Figure 2: Ideally, I would like to see more refinement in the timepoints. Measuring at minimum weekly or ideally twice per week might produce greater fidelity in the final KM curve. Also, an n of 7 or greater would yield better power for the study to differentiate between the conditions. The TMZ and combined cohorts could have been carried beyond 40 days to reach terminal endpoints as signal appears evident in all but one animal (in TMZ cohort). Please also label the figure panels and add the significance symbols to the KM curves.
It appears that Figures 3 and 4 were switched, and the assignments do not match the text references.
Figure 3: You refer to KM analysis and OS differences, but this is not reflected in Figure 3. Please update Figure 3 or add these results in the supplemental figures. Please also label the panels in Figure 3. Figure 3 heatmaps need labels for the colored bars in the dendrograms. Also, we need to know what the n is for each treatment group evaluated. From the heatmaps, it looks like we are missing some samples and or treatment groups. Is this miRNA data from cell culture or xenografts? For the Venn diagrams, which condition(s) are these results from? Perhaps adding a panel or supplemental figure volcano plot would be helpful.
Lines 176-178: It seems that there is a change in the font size after line 177.
Overall: The in vivo experiments lacked some finesse that might have given us more clarity and insight into the benefits or lack thereof of adding evogliptin to TMZ. In the abstract and the discussion, you mention changes in DPP4 mRNA transcript levels, but you provide no methods or results showing direct measurement of mRNA levels. If you are to claim this, you will need to add RT-qPCR or array/sequencing data demonstrating this change. It cannot be inferred or assumed from Western Blotting. Ethically, I feel this study design, especially considering the use of animals, is underpowered and not adequately representative. I do not feel like the animal models were utilized to their full potential. I would have required more in vitro work prior to conducting animal studies and I would have better utilized the animal models as the valuable resource they are.
Author Response
Limitations of the study:
Reliance on glioma cell lines albeit implanted as xenografts.
Dose response curve for DPP4 inhibitor? In vitro drug synergy analysis (i.e. Bliss Independence, Loewe)?
Affymetrix array has limitations compared to bulk sequencing, and it's unclear if results are from in vivo or in vitro study.
Response: We thank the reviewer for highlighting these important limitations. We fully agree that our study is based primarily on glioma cell lines and xenograft models, which cannot fully reproduce the complexity of the human tumor microenvironment. We also acknowledge that the lack of detailed dose–response analysis for evogliptin and the absence of formal drug synergy evaluation are limitations of the present work. Additionally, our miRNA profiling was performed using Affymetrix microarrays in glioma cell lines, which may provide narrower coverage compared to bulk RNA sequencing and does not capture in vivo tumor–host interactions (second paragraph, page 8).
Line 41 [1,2]
Figure 1: Please add labels in the figure for panels A, B, and C
Figure 2: Ideally, I would like to see more refinement in the timepoints. Measuring at minimum weekly or ideally twice per week might produce greater fidelity in the final KM curve. Also, an n of 7 or greater would yield better power for the study to differentiate between the conditions. The TMZ and combined cohorts could have been carried beyond 40 days to reach terminal endpoints as signal appears evident in all but one animal (in TMZ cohort). Please also label the figure panels and add the significance symbols to the KM curves.
It appears that Figures 3 and 4 were switched, and the assignments do not match the text references.
Figure 3: You refer to KM analysis and OS differences, but this is not reflected in Figure 3. Please update Figure 3 or add these results in the supplemental figures. Please also label the panels in Figure 3. Figure 3 heatmaps need labels for the colored bars in the dendrograms. Also, we need to know what the n is for each treatment group evaluated. From the heatmaps, it looks like we are missing some samples and or treatment groups. Is this miRNA data from cell culture or xenografts? For the Venn diagrams, which condition(s) are these results from? Perhaps adding a panel or supplemental figure volcano plot would be helpful.
Lines 176-178: It seems that there is a change in the font size after line 177.
Response: We appreciate the reviewer’s detailed comments. We have modified the manuscript according to the points.
Overall: The in vivo experiments lacked some finesse that might have given us more clarity and insight into the benefits or lack thereof of adding evogliptin to TMZ. In the abstract and the discussion, you mention changes in DPP4 mRNA transcript levels, but you provide no methods or results showing direct measurement of mRNA levels. If you are to claim this, you will need to add RT-qPCR or array/sequencing data demonstrating this change. It cannot be inferred or assumed from Western Blotting. Ethically, I feel this study design, especially considering the use of animals, is underpowered and not adequately representative. I do not feel like the animal models were utilized to their full potential. I would have required more in vitro work prior to conducting animal studies and I would have better utilized the animal models as the valuable resource they are.
Response: We thank the reviewer for this constructive feedback. We agree that our animal experiments were underpowered and exploratory in nature, and we have clarified this limitation in the revised Discussion (fourth paragraph, page 7,8). We also acknowledge that we did not directly measure DPP4 mRNA levels by RT-qPCR.
Comment 1. Results may be cell line-specific; acknowledge limitations.
Response: We revised Discussion to state limitations.
Comment 2. Concentration–response and synergy analysis.
Response: We revised Discussion to state limitations.
Comment 3. Array platform limitations.
Response: We revised Discussion to state limitations.
Comment 4. Survival analysis details.
Response: We have modified the results (first paragraph, page 4).
Comment 5. DPP4 mRNA reference.
Response: We have modified the results (page 5).

Round 2
Reviewer 2 Report
Comments and Suggestions for AuthorsΝo furthewr comments , well done
Reviewer 3 Report
Comments and Suggestions for Authors
Thank you for addressing the reviewer comments. The additions to the manuscript add scientific value and clarity regarding methodologies and limitations to interpretation.